# Exclusive Effect in Rydberg Atom-Based Multi-Band Microwave Communication

**Shuhang You** [1,2] **, Minghao Cai** [1,2,3,4]**, Haoan Zhang** [1,2]**, Zishan Xu** [1,2] **and Hongping Liu** [1,2,*]

1   State Key Laboratory of Magnetic Resonance and Atomic and Molecular Physics, WIPM, Innovation Academy for Precision Measurement Science and Technology, Chinese Academy of Sciences, Wuhan 430071, China
2   University of Chinese Academy of Sciences, Beijing 100049, China
3   Hefei National Laboratory, Hefei 230088, China
4   Shanghai Branch, Hefei National Laboratory, Shanghai 201315, China
*   Correspondence: liuhongping@wipm.ac.cn

**Abstract:** We have demonstrated a Rydberg atom-based two-band communication with the optically excited Rydberg state coupled to another pair of Rydberg states by two microwave fields, respectively. The initial Rydberg state is excited by a three-color electromagnetically-induced absorption in rubidium vapor cell via cascading transitions, with all of them located in infrared bands: a 780 nm laser servers as a probe to monitor the optical transmittancy via transition $5S_{1/2} \rightarrow 5P_{3/2}$, 776 nm and 1260 nm lasers are used to couple the states $5P_{3/2}$ and $5D_{5/2}$ and states $5D_{5/2}$ and $44F_{7/2}$. Experimentally, we show that two channel communications carried on the two microwave transitions influence each other irreconcilably, so that they cannot work at their most sensitive microwave-optical conversion points simultaneously. For a remarkable communication quality for both channels, the two microwave fields both have to make concessions to reach a common microwave-optical gain. The optimized balance for the two microwave intensities locates at $E_{MW1} = 6.5$ mV/cm and $E_{MW2} = 5.5$ mV/cm in our case. This mutual exclusive influence is theoretically well-explained by an optical Bloch equation considering all optical and microwave field interactions with atoms.

**Keywords:** Rydberg atom; electromagnetic induced absorption; Autler-Townes splitting

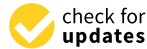



## 1. Introduction

As a magnificent combination between quantum science and engineering, quantum electric field intensity (EFI) sensing technology has aroused wide concern among scientists. In particular, Rydberg atom detection with ionization pulses is high efficient, but destructive [1], while a new means, called electromagnetically-induced transparency (EIT), has a unique feature of coherent non-destructive absorption of light [2,3]. This coherent advantage promotes the potential capacities of the quantum meter of EFI based on Rydberg atoms [4,5]. The EIT is a quantum interference phenomenon, in which the absorption of probe laser drops off sharply when an intense coupling laser causes the coherent destructive effect between two possible excitation pathways.

This Rydberg atom-based EFI sensing has the potential to replace the conventional metal dipole antenna-based technique as SI in the microwave(MW) frequency range by the virtue of traceability directly to the Planck constant $\hbar$ [6], as well as high sensitivity [7,8], compact system size, and a broad tuneability range from MHz to THz [9–13]. It has attracted a number of research towards the application in sensing and communication, such as engineering regarding the MW electric field intensity [7,8,14,15], phase [16], polarization [17,18], angle of incident [19], and miniaturization of system [6,20]. In addition to the absolute or vector field measurement, it has also made good progresses in subwavelength imaging [16,21].

With the development of quantum microwave E-field measurement, telecommunication based on the Rydberg atom has gradually exerted a tremendous fascination. Rydberg

atoms can revive the baseband signal modulated to the carrier MW resonant on the Rydberg energy levels, by the way of monitoring the probe laser transmission without demodulation, including amplitude modulation (AM) [22–24], frequency modulation (FM) [10,25,26], and phase modulation signals [16,27]. In addition, to realizing the broad and continuous tunable atom-based receiver, appending an adjacent resonance field [28] or Rydberg AC stark effect [29] is adopted successively.

To improve the data transfer capacity and efficiency, the multiband and multichannel methods works like a charm, which paves the way for concurrent telecommunication. Holloway et al. demonstrated the capture and recovery of the music analog signal by two different Rydberg atomic species in the same vapor cell, with double the number of lasers [30]. Song and Wang verified the feasibility of Rydberg atom-based frequency division multiplexing (FDM) via a resonant microwave and another nearby detuning one [31]. Jia et al. employed an auxiliary microwave resonant on adjacent Rydberg levels to significantly extend the Autler-Townes (AT) regime and improve its lower bound with several orders of magnitude in a Rb cell [32]. Recently, Du et al. realized concurrent two-channel analog and digital communications imposed on two different Rydberg states [33]. A more complex multi-channel configuration has also just been utilized by Cox et al. [34]. A deep learning model was also utilized in processing the multichannel signal of quantum telecommunication to promote its signal prediction capability and ability to identify information from noisy data without use of complex devices [35,36].

However, the concurrent multi-channel communication by modulating the baseband signals over multi-carrier MW resonant on Rydberg transitions is very complicated, since different Rydberg states are coupled with each other; thus, their MW-optical amplification gains are correlated [34]. In this paper, we employ two microwaves, respectively, resonant on contiguous $^{85}$Rb Rydberg transitions: $41F_{7/2} \leftrightarrow 41G_{9/2}$ ($\nu_{MW1} = 1.2$ GHz), $41F_{7/2} \leftrightarrow 42D_{5/2}$ ($\nu_{MW2} = 31.9$ GHz), serving as two bands, on each of which a baseband signal is modulated, 2 kHz and 4 kHz, as the channel signals. Therefore, two communication channels are built up based on diverse bands with enormous frequency differences. In this case, the different choice of band microwave power will affect the channel signal gains simultaneously. A deeper study of their correlation will help us to optimize the experimental condition best. Unlike the usual previous works [7,8,14,15], the Rydberg Rb atoms in our experiment are excited by three infrared lasers [37–39], and the signal is probed by electromagnetically-induced absorption (EIA), rather than EIT.

## 2. Experiment Setup

Our experimental setup is shown in Figure 1a, and the relevant energy levels of $^{85}$Rb diagram are depicted in Figure 1b. Similar to the kernel of the Rydberg atom-based receiver, our Rb atom vapor cell is shaped as $\Phi$25 mm $\times$ 70 mm cylindrical. We can see that the Rydberg Rb atoms are sequentially excited by three infrared lasers [37–39]. All the lights are generated from external cavity semiconductor lasers (Moglabs CEL series) and frequency-stabilized by Pound–Drever–Hall (PDH) system [40] with a ultra-stable Fabry–Perot cavity (UFPC). The combination of the half-wave plate ($\lambda/2-$WP) and polarization beam splitter (PBS) is able to finely adjust the power of the transmitted segment of laser, which takes part in the core experiment and another feeble reflected one entering into the PDH system. The remaining PBS is used to guarantee the polarization of lasers in the Rb cell and separate the 780 nm laser and 776 nm laser. Moreover, the 776 nm laser and 1260 nm lasers are combined efficiently via dichroic mirror (DM). Iris diaphragms (ID) are applied to reduce the stray light. The microwaves are generated by the two same types of MW sources (Anritsu 68369A) and then irradiated to the Rb cell through two rectangular horns, whose orientation is perpendicular to the lasers. Both MW generators are synchronized by a 10 MHz frequency reference provided by a common signal generator (RIGOL DG1022U).

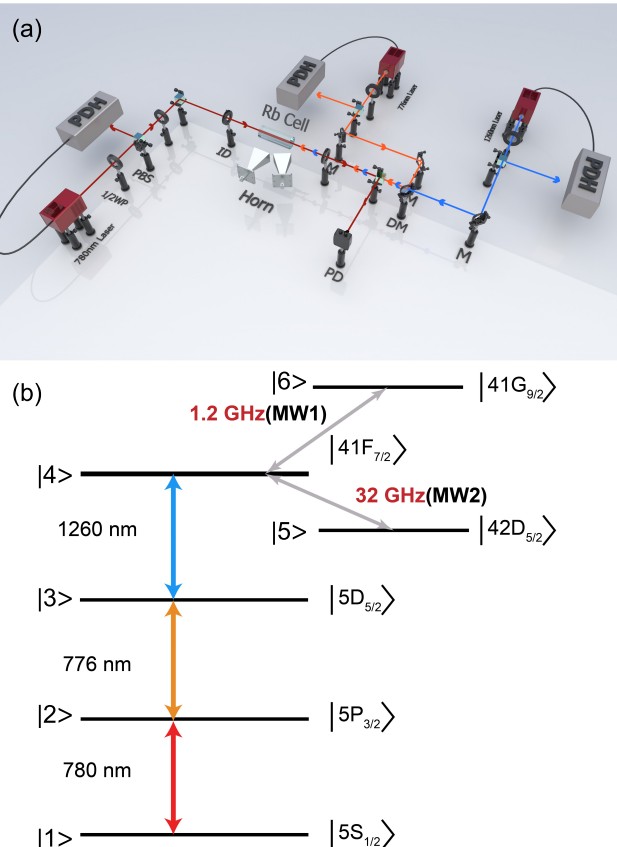

**Figure 1.** (**a**) Experimental schematic setup of the three infrared laser driven Rb Rydberg-based MW receiver and (**b**) relevant energy levels diagram. All infrared lights are generated by external cavity semiconductor lasers and frequency-stabilized by PDH system with the ultra stable Fabry–Perot cavity. The Rb atoms in vapor cell are excited successively to the $41F_{7/2}$ Rydberg state by 780 nm, 776 nm, 1260 nm lasers, which is also coupled with MW1 on the transition $41F_{7/2} \leftrightarrow 41G_{9/2}$ and MW2 on neighboring transition $41F_{7/2} \leftrightarrow 42D_{5/2}$, as shown in underlying figure. The terminology in the upper figure about optical components: $\lambda/2$-WP is acronym of a half-wave plate; PBS is polarization beam splitter; ID is iris diaphragms; PD is photodetector; DM is dichroic mirror and M is silver mirror.

As mentioned previously [37–39], the $^{85}$Rb atoms in vapor cell are excited successively to the $41F_{7/2}$ Rydberg state by three lasers, as shown in Figure 1b. The 780 nm laser is on resonance to $|5S_{1/2}, F = 3\rangle \leftrightarrow |5P_{3/2}, F = 4\rangle$ with a power of 500 μW and diameter of 2 mm, where an auxiliary Rb vapor is employed with its saturated absorption spectroscopy as the frequency reference. The 776 nm laser serves as the second coupling for two-step ladder EIT, with a power of 20 mW and a diameter of 2 mm. Finally, the 1260 nm laser subsequently excites atom to Rydberg state via three-step laser EIA with a power of 37 mW and a diameter of 2 mm and swept across the $|5D_{5/2}\rangle$ to $|41F_{7/2}\rangle$. It should be noted that the 776 nm laser is frequency-detuned for an optimal signal-noise-rate (SNR) of three-step laser EIA photoelectric signal, which has been studied in detail in our previous work [39]. All lasers are optionally locked to UFPC by PDH technique.

The microwave frequency is tuned to $\nu_{MW1} = 1.2$ GHz to drive the Rydberg transition of $^{85}$Rb $|41F_{7/2}\rangle \leftrightarrow |41G_{9/2}\rangle$ with a dipole moment $1637.6\ ea_0$, where $a_0$ is the Bohr radius and $e$ the elementary charge. The frequency of another MW is $\nu_{MW2} = 31.9$ GHz, driving the contiguous transition $|41F_{7/2}\rangle \leftrightarrow |42D_{5/2}\rangle$ with a dipole moment $1465.2\ ea_0$. The baseband signals are output from a signal generator, which has been synchronized with the one driving the two MW generators. They are imported into MW sources through its

external modulation port. We modulate the 2 kHz sinusoidal signal into MW1 and the 4 kHz into MW2, respectively. The modulation depth is 33% throughout the experiment.

### 3. Theory

In our work, a six-level model is employed to describe the atom–light and atom–microwave interactions, shown in Figure 1b. As adopted previously [39], the probe laser (780 nm) is resonant on the states $|1\rangle$ and $|2\rangle$, with Rabi frequency $\Omega_p$, the dressing laser (776 nm) detuned to the transition between states $|2\rangle$ and $|3\rangle$ with $\Omega_d$, and the coupling laser (1260 nm) scanning across the states $|3\rangle$ and $|4\rangle$ with $\Omega_c$. This forms a three-laser EIA configuration. Two microwave fields $\Omega_{MW1}$ and $\Omega_{MW2}$ are further applied to couple the Rydberg transitions $|4\rangle \leftrightarrow |5\rangle$ and $|4\rangle \leftrightarrow |6\rangle$, respectively, with its Rabi frequencies, expressed as

$$\Omega_i = \frac{E_i \cdot \mu_i}{\hbar} \quad (i = \mathrm{MW1}, \mathrm{MW2}), \tag{1}$$

where $E_i$ is the electric filed intensity of microwave and $\mu_i$ is its relevant Rydberg transition dipole moment. If we modulate a sinusoidal signal with angular frequency $\omega_i$ $(i = 1, 2)$ into the MW source generator by AM pattern under the modulation depth $\alpha$, we can define a new Rabi frequency as

$$\Omega'_i = \Omega_i(1 + \alpha \sin(\omega_i t)) \quad (i = \mathrm{MW1}, \mathrm{MW2}) \tag{2}$$

Therefore, the null atomic Hamiltonian of our system can be described as

$$
\begin{aligned}
H_0 = - \hbar [ & \Delta_{21}\sigma_{22} \\
& + (\Delta_{21} + \Delta_{32})\sigma_{33} \\
& + (\Delta_{21} + \Delta_{32} + \Delta_{43})\sigma_{44} \\
& + (\Delta_{21} + \Delta_{32} + \Delta_{43} - \Delta_{45})\sigma_{55} \\
& + (\Delta_{21} + \Delta_{32} + \Delta_{43} - \Delta_{45} + \Delta_{64})\sigma_{66} ]
\end{aligned} \tag{3}
$$

while the interaction Hamiltonian between atom and laser as

$$
\begin{aligned}
H_{\mathrm{L}} = \frac{\hbar}{2} [ & \Omega_p(\sigma_{12} + \sigma_{21}) + \Omega_d(\sigma_{23} + \sigma_{32}) \\
& + \Omega_c(\sigma_{34} + \sigma_{43}) ],
\end{aligned} \tag{4}
$$

where $\sigma_{ij} = |i\rangle\langle j|$ is atomic transition operators [41]. Additionally, the interaction Hamiltonian between atom and microwave is

$$H_{\mathrm{MW}} = \frac{\hbar}{2}[\Omega'_{\mathrm{MW1}}(\sigma_{45} + \sigma_{54}) + \Omega'_{\mathrm{MW2}}(\sigma_{46} + \sigma_{64})]. \tag{5}$$

Thus, the total Hamiltonian of six-level system is $H = H_0 + H_{\mathrm{L}} + H_{\mathrm{MW}}$ and, in the rotating-wave approximation, can be rewritten as

$$
H = -\frac{\hbar}{2}
\begin{pmatrix}
0 & \Omega_p & 0 & 0 & 0 & 0 \\
\Omega_p & 2\Delta_2 & \Omega_d & 0 & 0 & 0 \\
0 & \Omega_d & 2\Delta_3 & \Omega_c & 0 & 0 \\
0 & 0 & \Omega_c & 2\Delta_4 & \Omega'_{\mathrm{MW1}} & \Omega'_{\mathrm{MW2}} \\
0 & 0 & 0 & \Omega'_{\mathrm{MW1}} & 2\Delta_5 & 0 \\
0 & 0 & 0 & \Omega'_{\mathrm{MW2}} & 0 & 2\Delta_6
\end{pmatrix}, \tag{6}
$$

where $\Delta_2 = \Delta_{21}$, $\Delta_3 = \Delta_{21} + \Delta_{32}$, $\Delta_4 = \Delta_{21} + \Delta_{32} + \Delta_{43}$, $\Delta_5 = \Delta_{21} + \Delta_{32} + \Delta_{43} - \Delta_{45}$, $\Delta_6 = \Delta_{21} + \Delta_{32} + \Delta_{43} - \Delta_{45} + \Delta_{64}$.

Hence, the time evolution of the whole system can be forecasted by a Bloch equation with Lindblad decay term,

$$\dot{\rho} = -\frac{i}{\hbar}[H, \rho] + \mathcal{L}_\gamma \rho \tag{7}$$

where $\rho$ is the density operator and $\mathcal{L}_\gamma \rho$ the standard Lindblad decay term

$$\mathcal{L}_\gamma \rho = \sum_i \Gamma_i [L_i \rho(t) L_i^\dagger - \frac{1}{2}\{L_i^\dagger L_i, \rho(t)\}], \tag{8}$$

where $\Gamma_i$ is the decay rate of the energy levels and $L_i$ is the jump operator while $L_i^\dagger$ is its conjugation. We numerically solve Equation (7) to find the steady-state solution for $\rho_{12}$, representing the absorption of the probe laser for various $\Omega_{MW1}$ and $\Omega_{MW2}$. To take the Doppler effect into consideration, a further integration is performed for $\rho_{12}(\Omega'_{MW1}, \Omega'_{MW2})$ [42],

$$\rho'_{12} = \frac{1}{\sqrt{\pi}u} \int_{-\infty}^{\infty} \rho_{12}(\Delta'_{12}, \Delta'_{23}, \Delta'_{34}) e^{-v^2/u^2} dv \tag{9}$$

where $v$ is the velocity of atoms, $u = \sqrt{2k_B T/M}$ is the most probable speed of atom determined by Boltzmann constant, temperature $T$, and atomic mass $M$, while $\Delta'_{12}$, $\Delta'_{23}$, $\Delta'_{34}$ are the modified detuning of lasers, as follows:

$$
\begin{aligned}
\Delta'_{12} &= \Delta_{12} - \frac{2\pi}{\lambda_p}v \\
\Delta'_{23} &= \Delta_{23} - \frac{2\pi}{\lambda_d}v \\
\Delta'_{34} &= \Delta_{34} - \frac{2\pi}{\lambda_c}v
\end{aligned}
\tag{10}
$$

## 4. Result and Discussion

The experiment is configured as previously [39], where the standard sinusoidal waveform, as the baseband signal, is modulated into the high-frequency MW resonant on Rydberg transition, such as 1 kHz, over the 1.2 GHz MW. The 1 kHz wave would be restored by the self-demodulation effect on Rydberg Rb atoms. Rather than modulating the baseband signal over single-carrier MW [8], here we performed a concurrent multi-channel communication by modulating the baseband signals over multi-carrier MWs resonant on different Rydberg transitions. In the latter case, different Rydberg states are coupled each other; thus, their MW-optical amplification gains are correlated each other. In our experiment, as shown in Figure 1, we built up two communication channels on diverse bands by employing two microwaves: one low-frequency $\nu_{MW1} = 1.2$ GHz and one high-frequency $\nu_{MW2} = 31.9$ GHz, which are resonant on transitions to a pair of adjacent Rydberg levels. The two frequency-different baseband signals, 2 kHz and 4 kHz, are modulated onto them, respectively, as the targets.

We can calibrate the MW E-intensity sensed by Rydberg atoms via the standard antenna method [39]. The calibration is performed at a strong enough MW E-intensity for a better linearity for the EIA spectral loss [39,43]. It has a value of $256.3 \, \text{mV} \cdot \text{cm}^{-1}/\sqrt{\text{W}}$ for MW1 and $1360.1 \, \text{mV} \cdot \text{cm}^{-1}/\sqrt{\text{W}}$ for MW2. The frequency bandwidth of the baseband signal is 20 kHz for channel 1.2 GHz and 60 kHz for channel 31.9 GHz, but both of them have remarkable gains, within 1~5 kHz, where we can investigate the multi-band communication as mentioned in the previous paragraph. Here, the dressing laser is also locked at the detuning point $\Delta_d = 2\pi \times (-6.36)$ MHz by PDH technique, as in the previous work [39], to avoid noise induced by pump effect [44].

In the concurrent communication, both of the carrier microwave fields couple to a common Rydberg state, which might lead to the mutual influence of the baseband signal gains. For simplicity, we study the dependence of optical signal variation in one channel at its carrier microwave power on the other channel carrier microwave power. This signal

variation is recorded by the optical sinusoidal signal amplitude driven at some certain microwave sinusoidal baseband input. It indicates the microwave-optical conversion gain. It is shown in Figure 2. Here, the Rydberg state 6 (41G$_{9/2}$) and state 5 (42D$_{5/2}$) couple to the common Rydberg state 4 (41F$_{7/2}$), as depicted in Figure 1.

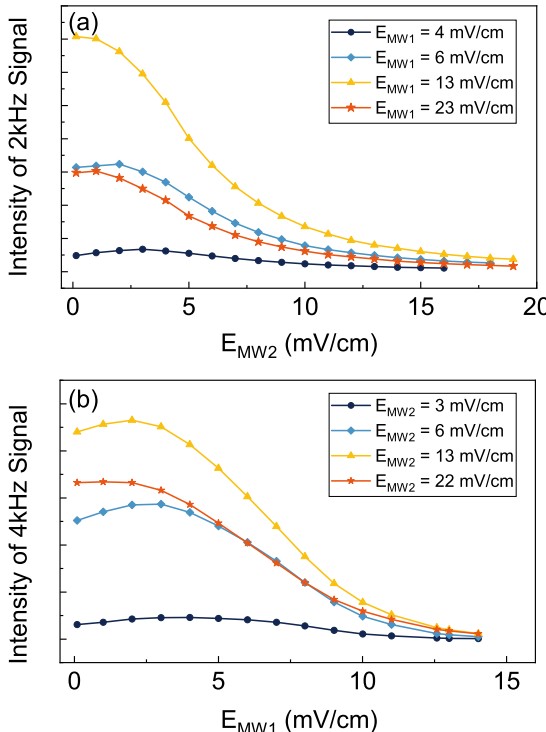

**Figure 2.** The recorded photoelectric signal intensity of baseband signals by our Rydberg atom-based receiver for different E-intensities of MW1 and MW2. In either case, (**a**) or (**b**), the intensity of the baseband signal with $\omega_1 = 2 \times 2\pi$ kHz ($\omega_2 = 4 \times 2\pi$ kHz) modulated onto MW1 (MW2) gradually decay with the intrusion of the opponent MW power beyond $E_{MW} \sim 2.5$ mV/cm. However, in both cases, the signal intensity climbs up along its own channel microwave intensity until reaching the maximum at $E_{MW} \sim 13$ mV/cm, and then drops again, implying an available power optimization.

As an example, it can be seen from Figure 2a that the $\omega_1 = 2 \times 2\pi$ kHz sinusoidal signal intensity in carrier channel MW1 slightly climbs up and then drops down with the microwave power increasing of MW2 at a given microwave power of the other channel MW1. It has a maximum value at $E_{MW1} = 13$ mV/cm, hinting a highest microwave-optical gain, while it drops at other MW powers, such as $E_{MW1} = 4, 6, 23$ mV/cm. However, all of them are going to attenuate along with the intrusion of the other microwave coupling in channel MW2. The stronger the microwave power MW2 applied, the smaller the gain gets for MW1, especially for the highest gain curve at $E_{MW1} = 13$ mV/cm. This is because the AT splitting interval of Rydberg EIA spectroscopy induced by MW1 is mildly widened, due to the participation of MW2, which drives the neighboring transition [32]. A simple explanation is that the driving causes the common Rydberg state level to shift and subsequently leads to the AT splitting change, and finally, the microwave-optical amplification in MW1 is weakened. This phenomenon is more obvious in Figure 2b, where we investigate the influence of MW1 on channel MW2. Here, the $\omega_2 = 4 \times 2\pi$ kHz sinusoidal signal is coupled through the carrier channel MW2 alone. The interesting thing is that both have the best optimized microwave-optical gains at $E_{MW} = 13$ mV/cm [8]. This is due to the fact that the two Rydberg state couplings own close electric dipole moments. In the same microwave fields, they will contribute close amplitudes of interaction in the Hamiltonian shown in Equation (6).

This mutual influence can also been investigated by supervising two different baseband optical signals simultaneously. In this case, the optical signal on the photoelectric detector is the weighting superposition of 2 kHz and 4 kHz sinusoidal curves. A fast Fourier transform (FFT) is applied to the recorded signal to pick up the two frequency components, as shown in Figure 3. It displays the power spectrum of FFT of the photoelectric signal accepted by our Rydberg atom-based receiver in the condition of various MW E-intensity. From the perspective of the Figure 3a, the intensity of 2 kHz component firstly increases then decreases as the E-intensity of MW1 goes up because the optical gain variation characteristic is parabolic. It has a maximum around $E_{MW1} = 11$ mV/cm at fixed $E_{MW2} = 7$ mV/cm. However, although the E-intensity of MW2 is fixed, the intensity of 4 kHz component gradually declines monotonically, along with the MW1 power increasing. This reveals that the application of the other carrier channel MW1 always plays a negative role for the microwave-optical amplification in channel MW2. The same conclusion can also be drawn from Figure 3b for the influence of channel MW1 on signal 2. In a word, the intensity of 2 kHz and 4 kHz components all suffer from the opponent carrier power, namely MW2 and MW1, respectively, here.

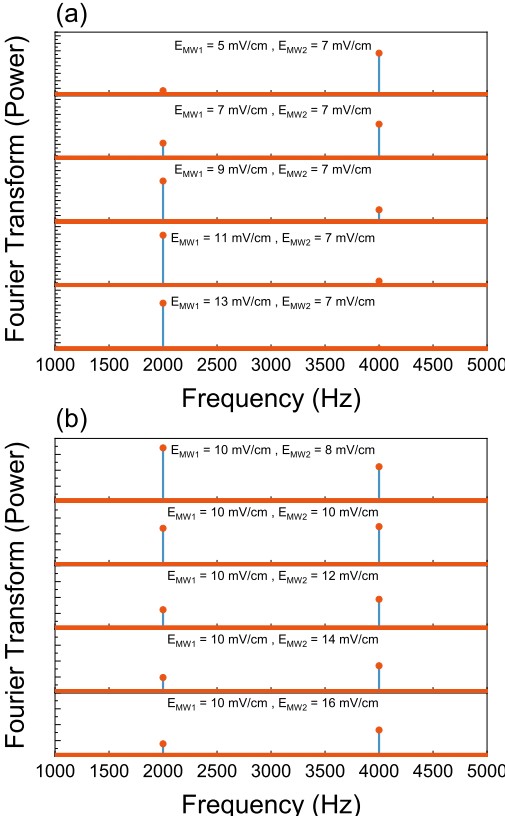

**Figure 3.** The power spectrum of fast Fourier transform of the photoelectric signal accepted by the Rydberg atom-based receiver in the condition of various pairs of MW E-intensities. (**a**) The increasing of the MW1 power enhances the signal amplitude of its own channel, until $E_{MW1} \sim 11$ mV/cm, but weakens its opponent always gradually. (**b**) Similar to (**a**), but the turning point occurs at $E_{MW2} \sim 10$ mV/cm.

This malicious damage to each other leads to a situation in which both channels cannot work at their own optimized MW power points for simultaneously satisfying the microwave-optical gains, which is neglected in previous work [33]. For an acceptable dual channel communication, we have to negotiate for these parameters, for example, $E_{MW1} = 10$ mV/cm and $E_{MW2} = 10$ mV/cm is a good pair of power parameters in our case, of which, a conclusion is drawn from Figure 3b, although the data points are not enough for an accurate optimization.

Therefore, the microwave-optical amplification gains are the functions of the two applied carrier microwave powers, and a full and complete data surface in these two-dimension parameter spaces should be obtained before performing a balanced dual channel communication. Experimentally, we can collect the photoelectric signals received by the Rydberg atom-based sensor in step-by-step variations of MW1 and MW2, as shown in Figure 4a. For either the 2 kHz or 4 kHz channels, the introduction of the other MW power leads to the microwave-optical gain decreasing, as represented by the gradual pink surface and the gradual dark blue surface. The two surfaces intersect each other, sharing a common gain line in magenta, on which two channels have the same gain coefficient. The optimized MW powers locate at the highest intensity point on this line. It is $E_{MW1} = 6.5$ mV/cm and $E_{MW2} = 5.5$ mV/cm, more accurate than the estimation from Figure 3b.

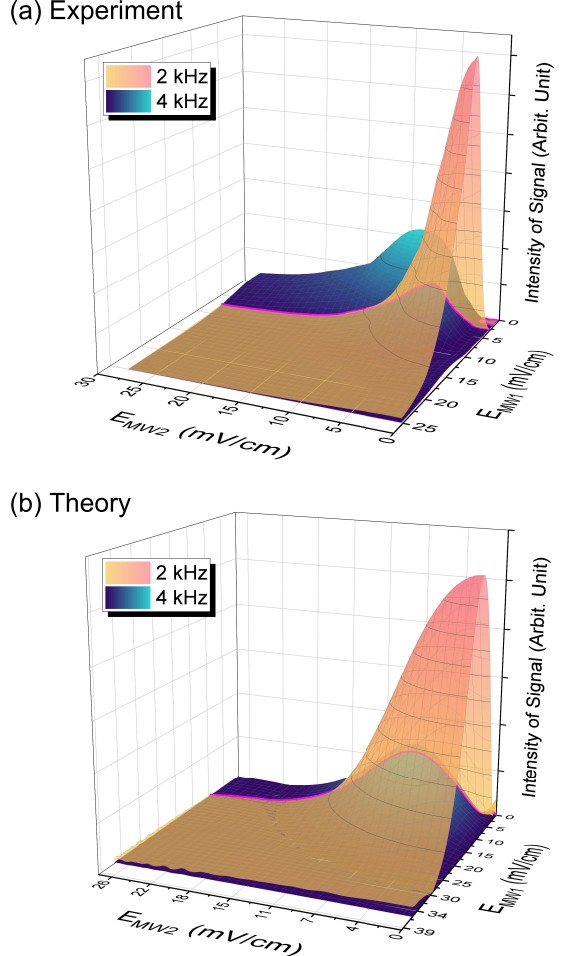

**Figure 4.** (**a**) Experimental and (**b**) theoretical result of the situation that the power of two channels are changed simultaneously. The gradual pink surface represents the variation of $\omega_1 = 2 \times 2\pi$ kHz frequency component, and the gradual dark blue surface signifies the $\omega_2 = 4 \times 2\pi$ kHz frequency component. The magenta line is the intersection of two surfaces, which means the intensity of two frequency component are equal when the work point is set on this line. Whether from the experimental observation or the theoretical simulation, we can see that the influence of one channel on the optical gain reduction of the other channel is very obvious, and the theoretical and experimental results are consistent in the trend.

It can also be noticed that the 2 kHz channel (MW1) sacrifices itself mostly to keep the same microwave-optical gain, since its maximum is three times larger than that of 4 kHz channel (MW2) if it works alone. This experimental observation is well-explained

by a theoretical simulation. It is shown in Figure 4b. The discrepancy might come from the inaccuracy of the optical parameters used in the simulation. As described in the theory section, the optical absorption is proportional to the imaginary part of susceptibility averaged over a Maxwell–Boltzmann velocity distribution at room temperature. It is parameterized by the laser wavelength detunings and laser intensities, as well as microwave coupling strength. The baseband signal is introduced into the microwave coupling by Equation (2). Thus, we can obtain the imaginary part of $\rho_{12}(\Omega'_{MW1}, \Omega'_{MW2})$ with evolution time. Subsequently, a fast Fourier transformation gives the intensities of the two frequency components at different carrier microwave Rabi frequencies, as presented in Figure 4b. It can be seen that the magenta line is the intersection of two surface, which means the intensity of two frequency component are equal when the work point is set on this line.

This mutual exclusive influence on each other channels might be easy to confuse with the electrometric field measurement span enhanced with an auxiliary microwave field coupling the optically excited Rydberg state to another neighbor one [32]. There, when a target electric field is very weak, it cannot induce remarkable AT splitting only where the spectral splitting is well-resolved. Rather than directly and strictly correlated with the electric field magnitude by the formula in Equation (1), it is described by a modified formula with the auxiliary field strength as an additional parameter [32].

However, although the auxiliary microwave supplies drive for the forming of the obvious AT splitting, the probe optical response is reduced greatly. A theoretical simulation is shown in Figure 5a, where the target channel (MW1) has a microwave strength with remarkable AT splitting, but with an auxiliary channel (MW2) with no and a weak microwave applied. The little change of microwave strength is only $\Delta\Omega_{MW2} = 0.5 \times 2\pi$ MHz, corresponding to little difference for the AT splitting. It implies a very small dynamic microwave-optical amplifying gain for the target signal. However, on the contrary, the gain has magnitude for the auxiliary field itself. The simulation is also presented, as shown in Figure 5b, where the AT splitting is obviously spanned, compared with the case in Figure 5a [32]. The change of microwave strength is set to the same magnitude $\Delta\Omega_{MW1} = 0.5 \times 2\pi$ MHz. To observe the spectral change, the Doppler broadening is not included in the calculation.

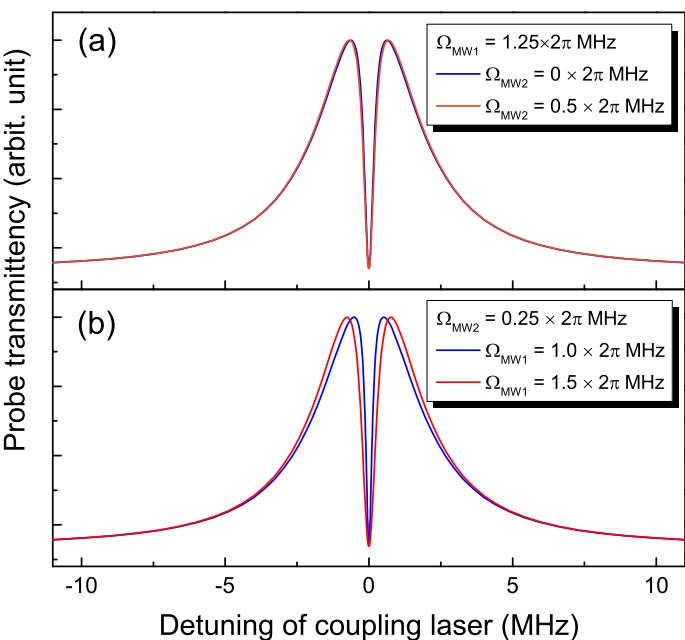

**Figure 5.** The spectral AT splitting for small change of the microwave field at target signal channel (**a**) and auxiliary channel (**b**). The auxiliary field helps to make the AT splitting gets more obvious for better quantization of the weak target field, but it reduces the sensitivity of the dynamic microwave-optical amplifying gain, which is much smaller than that of the auxiliary field itself.

## 5. Conclusions

In conclusion, we have experimentally and theoretically investigated two-band communication with the optically excited Rydberg state shared by two microwave fields coupling to another pair of Rydberg states. For the case with only one microwave channel, we can optimize the dynamic microwave-optical conversion gain to the maximum by selecting an appropriate strength for the carrier microwave field. However, this gain value is weakened by the intrusion of another microwave channel when a two-band communication is performed. We have to optimize the carrier microwave field strength again. To reach a common microwave-optical gain for the two-band communication, both carrier microwave field intensities have to be optimized. In our case, their values are found to locate at $E_{MW1} = 6.5$ mV/cm and $E_{MW2} = 5.5$ mV/cm, respectively. This optimization provides a mechanism to perform the communication when the signals are modulated into the atomic system through different carrier microwave fields.

**Author Contributions:** Conceptualization, H.L.; methodology, H.L.; software, H.Z.; validation, H.L.; formal analysis, M.C.; investigation, S.Y.; data curation, S.Y.; writing—original draft preparation, S.Y.; supervision, Z.X.; writing—review and editing, H.L.; project administration, H.L.; funding acquisition, H.L. All authors have read and agreed to the published version of the manuscript.

**Funding:** This research was supported by the National Key Research and Development Program of China (No. 2021YFF0603704), the National Natural Science Foundation of China (under Grants No. 12074388 and No. 12004393), and the Innovation Program for Quantum Science and Technology (No. 2021ZD0302100).

**Institutional Review Board Statement:** Not applicable.

**Informed Consent Statement:** Not applicable.

**Data Availability Statement:** Not applicable.

**Conflicts of Interest:** The authors declare no conflict of interest.

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
