# Peer review of "Exclusive Effect in Rydberg Atom-Based Multi-Band Microwave Communication"

_photonics, doi:10.3390/photonics10030328_

Round 1

Reviewer 1 Report

The authors studied two-band communication with the optically excited Rydberg state shared by two microwave fields coupling to another pair of Rydberg states via EIT. The experimental results can be explained correctly by theoretical way with optical Bloch equation considering all optical and microwave field interaction with atom. This work is very interesting and the result is reliable. Above all, I support its publication in Photonics.

Author Response

Thank you for your approval! I'm so excited to recieve your comments. It is inspired.

Reviewer 2 Report

In this paper, authors have experimentally and theoretically investigated two-band communication with the optically excited Rydberg state shared by two microwave fields coupling to another pair of Rydberg states. Two channel communications carried on the two microwave transitions influence each other irreconcilably and they can’t work at their most sensitive microwave-optical conversion spots simultaneously. The two microwave fields both have to make concessions to reach a common microwave-optical gain. This mutual exclusive influence is well theoretically explained by an optical Bloch equation considering all optical and microwave field interaction with atom. It is interesting and is of great application value. So, I can recommend it to be publicated on Photonics.

Author Response

(The authors gave the same response as above.)

Reviewer 3 Report

This manuscript reports the experimental studies of the mutual exclusive influence in two-band communication using the Rydberg states under microwave excitation. It shows that even with significant frequency differences, two-channel communication is not able to work when both microwaves are at their optimal gains and a common microwave-optical gain must be reached for better communication effect. The general illustration for the experiment and result is acceptable, but I would recommend reconsidering this manuscript after major revision due to the following reasons.

1.      Too many grammar errors in the abstract, introduction, and conclusion sections. Please consider re-writing these three parts and proofreading them multiple times. The current draft is very hard for the readers to understand.

2.      In the experiment setup part, please consider providing more details that how the experiment was done and what kind of data you collected from your apparatus. The authors mentioned that the experiment is configured as previously but I think it is worth illustrating them clearly for readers to better understand it.

3.      In Figure 4(b), the authors show theoretical results and compared them with the experimental results. However, I did not find a detailed discussion in the manuscript that how the theoretical results were done. Please consider adding more details.

Author Response

Dear editor and referees:

Thank the referees for the comments and we have improved the manuscript according the suggestions. All the replies to the questions are presented one by one, which are also marked in the main text of the updated manuscript. 

Best wishes.

Point 1: Too many grammar errors in the abstract, introduction, and conclusion sections. Please consider re-writing these three parts and proofreading them multiple times. The current draft is very hard for the readers to understand.

Response 1: We have proofreaded the three section and made many modifications. Thank you for your suggestion.

Point 2:  In the experiment setup part, please consider providing more details that how the experiment was done and what kind of data you collected from your apparatus. The authors mentioned that the experiment is configured as previously but I think it is worth illustrating them clearly for readers to better understand it.

Response 2: We have illustared breifly the previous work content as marked in uploaded manuscript.

Point 3:  In Figure 4(b), the authors show theoretical results and compared them with the experimental results. However, I did not find a detailed discussion in the manuscript that how the theoretical results were done. Please consider adding more details.

Response 3: It’s indeed true and we should have provided more details about our theoretical process. Now we have added the relevant content in the uploaded manuscript so that readers can get how it works. Thank you for your suggestion.

Author Response

Dear editor and referees:

Thank the referees for the comments and we have improved the manuscript according the suggestions. All the replies to the questions are presented one by one, which are also marked in the main text of the updated manuscript. 

Best wishes.

Point 1: Line 81, “Iris diaphragms (ID) are applied to shape lasers.” →“Iris diaphragms (ID) are applied to reduce the stray light.”

Response 1: Thanks. We have corrected this sentence according to the suggection.

Point 2:  Line 88, “with a power of 500 ¯W and”, mW(?).

Response 2: It has been corrected.

Point 3:  In Fig. 1, there are couple components without labeling, component after 776nm laser, component an 780nm laser after PBS.

Response 3: The component is 1/2 wave plate and acctually it has been labeled with only one example in Fig.1 for space-limiting.

Point 4:  Line 164, “common Rydberg state 4 (41F1/2) as” “common Rydberg state 4 (41F7/2) as”.

Response 4: It has been corrected.

Point 5:  Fig. 5, the vertical axis on (a) experiment and (b) theory should be in “Arbit. Unit”.

Response 5: We have corrected it.

Point 6:  “can’t” in the manuscript is better use “cannot”.

Response 6: We have taken all “can’t” instead of “cannot”. Thank you for your suggestion.

Round 2

Reviewer 3 Report

The authors made improvements to the manuscript and have provided the missing information that I asked for. I would like to recommend accepting this manuscript after further proofreading to polish the language.